# TLR3 activation mediates partial epithelial-to-mesenchymal transition in human keratinocytes

Andrea M Schneider[1], Robert P Feehan[1], Mackenzie L Sennett[1], Carson A Wills[2], Charlotte Garner[1],
Zhaoyuan Cong[1], Elizabeth M Billingsley[1], Alexandra F Flamm[1], Lisa M Shantz[1,3], Amanda M Nelson[1]

**TLR3 is expressed in human skin and keratinocytes, and given its varied role in skin inflammation, development, and regeneration, we sought to determine the cellular response in normal human keratinocytes to TLR3 activation. We investigated this mechanism by treating primary human keratinocytes with both UVB, an endogenous and physiologic TLR3 activator, and poly(I:C), a synthetic and selective TLR3 ligand. TLR3 activation with either UVB or poly(I:C) altered keratinocyte morphology, coinciding with the key features of epithelial-to-mesenchymal transition: increased epithelial-to-mesenchymal transition gene expression, enhanced migration, and increased invasion properties. These results confirm and extend previous studies demonstrating that in addition to its classical role in the innate immune response, TLR3 signaling also regulates stem cell–like properties and developmental programs.**

## Introduction

TLR3, a pattern recognition receptor activated by dsRNA, induces the expression of inflammatory cytokines and IFNs through NF-κB and interferon regulatory factor 3 activation (Kawai & Akira, 2010). TLR3 was originally described as a key regulator of dorsal–ventral patterning during embryogenesis and development in Drosophila (Anderson et al, 1985). TLR3 signaling has since been found to contribute to induced pluripotent stem cell development, hair follicle morphogenesis, skin regeneration, and stem cell mobilization through activation of important developmental pathways controlled by Wnt, Shh, and NF-κB (Lee et al, 2012; Mastri et al, 2012; Jia et al, 2015; Nelson et al, 2015). Of particular interest to the studies described here, TLR3 senses UVB-induced dsRNA that is released from UVB-damaged keratinocytes, resulting in skin inflammation (Bernard et al, 2012).

Because TLR3 is expressed in human skin and keratinocytes and given its varied role in skin inflammation, development, and regeneration, we sought to determine the cellular response in normal human keratinocytes to TLR3 activation. UVB initiates TNF-α and IL-6–mediated skin inflammation through TLR3 (Bernard et al, 2012); yet, the underlying molecular mechanisms of how UVB-mediated TLR3 signaling impacts keratinocyte physiology are not well defined. We investigated this mechanism by treating primary human keratinocytes with both UVB, an endogenous and physiologic TLR3 activator, and poly(I:C), a synthetic and selective TLR3 ligand, which may also stimulate RIG-I and MDA5 signaling in some experimental contexts (Kato et al, 2006; Nakata et al, 2021). TLR3 activation with either UVB or poly(I:C) altered keratinocyte morphology, coinciding with the key features of epithelial-to-mesenchymal transition (EMT): increased EMT gene expression, enhanced migration, and increased invasion properties. These results confirm and extend previous studies demonstrating that in addition to its classical role in the innate immune response, TLR3 signaling also regulates stem cell–like properties and developmental programs (Belvin & Anderson, 1996; Lee et al, 2012; Nelson et al, 2015).

## Results

### TLR3 activation triggers morphological change in normal keratinocytes

We noted a striking change in keratinocyte morphology after treatment with either UVB or poly(I:C). Keratinocytes lost their characteristic cuboidal, cobblestone shape and adopted an elongated spindle-like shape, very similar to fibroblasts. Keratinocytes exhibited this fibroblast-like morphological change at 48 h, which became more robust and sustained by 72 h (Fig 1A). This phenotype was observed after UVB (10 mJ/cm$^2$) exposure or poly(I:C) treatment (20 μg/ml) (Fig 1B and C). Approximately 35% of cells developed this spindle-like morphology by 72 h with poly(I:C)

[1]Department of Dermatology, Penn State Health Hershey Medical Center, Hershey, PA, USA   [2]Department of Pediatrics, Penn State College of Medicine, Hershey, PA, USA
[3]Department of Cellular and Molecular Physiology, Penn State College of Medicine, Hershey, PA, USA

Correspondence: anelson@pennstatehealth.psu.edu
Carson A Wills's present address is Department of Internal Medicine, Hospital of the University of Pennsylvania, Philadelphia, PA, USA
Alexandra F Flamm's present address is Department of Dermatology, NYU Grossman School of Medicine, New York, NY, USA

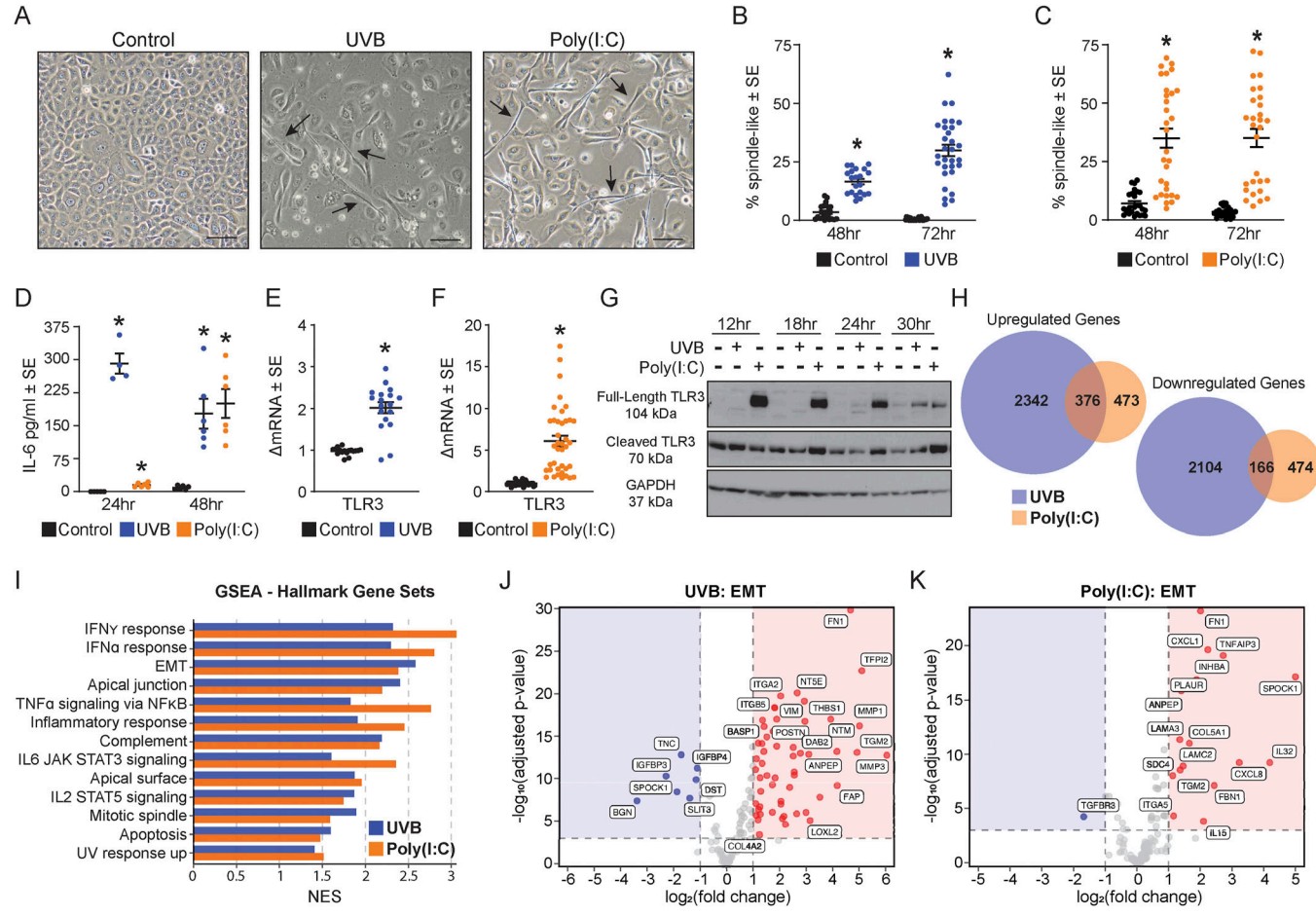

**Figure 1. TLR3 activation triggers morphological change in normal keratinocytes.**
**(A)** Representative images of normal human epidermal keratinocytes 72 h after UVB (10 mJ/cm²) exposure or poly(I:C) (20 µg/ml) treatment. Cells exposed to UVB are given fresh media 24 h after exposure and left in culture for another 48 h; poly(I:C) is on the cells during the first 24 h, followed by a washout period of 48 h (72 h in total). Images were captured at 10X magnification; scale bar represents 100 µm. **(B)** Quantification of spindle-like cells 48 and 72 h after UVB (10 mJ/cm²) exposure, n ≥ 21 images quantified. **(C)** Quantification of spindle-like cells 48 and 72 h after poly(I:C) (20 µg/ml), n ≥ 24 images quantified. **(D)** IL-6 ELISA data from the media of keratinocytes exposed to UVB (10 mJ/cm²) or treated with poly(I:C) (20 µg/ml) 24 and 48 h after treatment, unpaired *t* test, two-tailed, *P* < 0.05, n ≥ 4. **(E)** *TLR3* gene expression by qRT–PCR 48 h after UVB (10 mJ/cm²). Data were normalized to *RPLP0*, n ≥ 16. **(F)** *TLR3* gene expression by qRT–PCR 48 h after poly(I:C) (20 µg/ml). Data were normalized to *RPLP0*, n ≥ 36. **(G)** Representative immunoblot from keratinocytes exposed to UVB (10 mJ/cm²) and poly(I:C) (20 µg/ml) shows increases in TLR3 protein (full-length and cleaved) by immunoblot, n = 3. **(H)** Number of differentially expressed genes in 72-h UVB-treated cells and poly(I:C)-treated cells and their overlap. The number of overlapping genes is significant using Fisher's exact test, *P* = 0. **(I)** Hallmark gene sets of interest. NES, normalized enrichment score. **(J, K)** Volcano plots illustrating changes in the genes involved in epithelial-to-mesenchymal transition after (J) UVB and (K) poly(I:C) treatment. * denotes significance compared with the control, *P* < 0.05, Mann–Whitney *t* test, two-tailed, unless otherwise noted above.

treatment, whereas 30% of cells had undergone a similar morphological change 72 h after UVB exposure (Fig 1B and C). We confirmed UVB- and poly(I:C)-mediated activation of TLR3 in keratinocytes by increased production of IL-6, a cytokine downstream of TLR3 activation, and by up-regulation of both *TLR3* mRNA and protein levels (Fig 1D–G). TLR3 is unique in that its protein expression is up-regulated by its own activation, which then leads to an accumulation of the cleaved active form of TLR3 (Fig 1G) (Garcia-Cattaneo et al, 2012).

The parallels in morphological changes in UVB- and poly(I:C)-treated keratinocytes suggested that these two stimuli activate common pathways that result in the morphological changes we observed. We used RNA sequencing (RNA-seq) to gain a broad understanding of the genes and pathways impacted by UVB and

poly(I:C) treatment in keratinocytes. Cells were exposed to 10 mJ/cm² UVB or 20 µg/ml poly(I:C) for 24 h and harvested at 72 h after treatment initiation. Differential gene expression identified a total of 4,988 genes that were changed with UVB exposure (2,718 up-regulated and 2,270 down-regulated; adjusted *P* < 0.05), and 1,489 genes that were differentially expressed with poly(I:C) treatment (849 up-regulated and 640 down-regulated; adjusted *P* < 0.05). We observed a significant overlap in the differentially expressed genes between UVB and poly(I:C) treatment in normal keratinocytes: 376 genes increased and 166 genes decreased in both conditions, suggesting that these two stimuli activate common pathways in keratinocytes (Fig 1H).

Gene set enrichment analysis identified significantly enriched hallmark pathways common between UVB and poly(I:C) treatment,

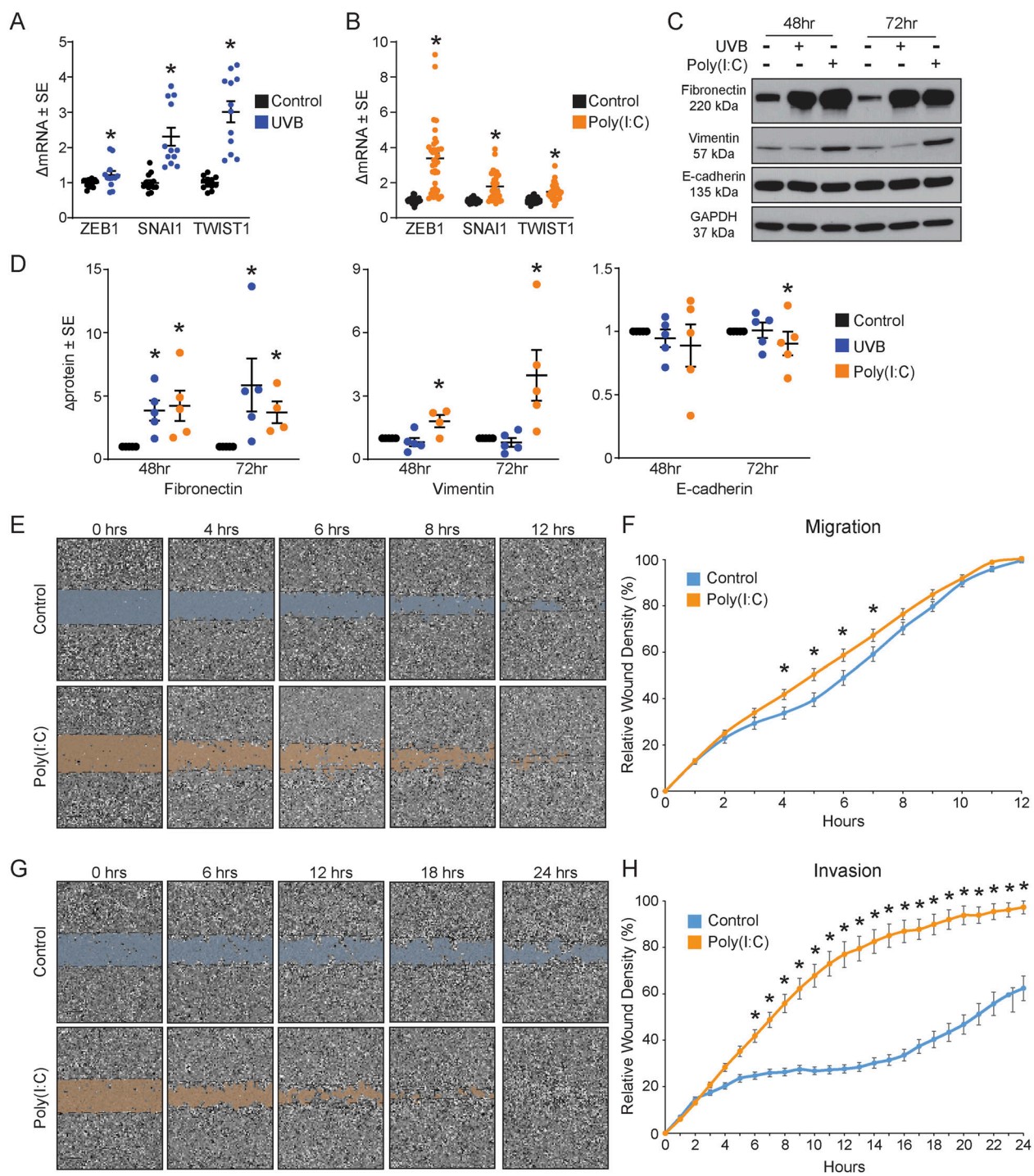

**Figure 2. TLR3 activation increases key epithelial-to-mesenchymal transition (EMT) markers and confers migration and invasion properties on keratinocytes.**
**(A)** Gene expression of EMT-associated transcription factors, *ZEB1*, *SNAI1*, and *TWIST1*, in keratinocytes treated with 10 mJ/cm$^2$ UVB by qRT–PCR 72 h after exposure, normalized to *RPLP0*, Mann–Whitney *t* test, one-tailed, *P* < 0.05, n = 12. **(B)** Gene expression of EMT-associated transcription factors, *ZEB1*, *SNAI1*, and *TWIST1*, in keratinocytes treated with 20 μg/ml poly(I:C) by qRT–PCR 72 h after exposure, normalized to *RPLP0*, Mann–Whitney *t* test, one-tailed, *P* < 0.05, n ≥ 30. **(C)** Protein levels of mesenchymal proteins (fibronectin and vimentin) and epithelial protein (E-cadherin) analyzed at 48 and 72 h after 10 mJ/cm$^2$ UVB or 20 μg/ml poly(I:C) exposure by Western blot, representative image. **(D)** Protein level after UVB treatment (10 mJ/cm$^2$) or poly(I:C) (20 μg/ml) was semi-quantified by densitometric analysis. Samples were normalized to GAPDH, and then, fold change was calculated compared with the control, paired *t* test, one-tailed, *P* < 0.05, n ≥ 4. **(E)** Representative images of the migration assay after scratches at 0 h, 10x magnification; the scale bar represents 200 μm, n ≥ 5 technical replicate wells per group. **(F)** Graphical representation of relative wound density overtime for control and poly(I:C)-treated cells, n ≥ 29 total wells. Migration data were analyzed by multiple *t* tests, one per time-point, using the Holm–Sidak correction method, with alpha = 0.05, and a consistent SD was assumed. **(G)** Representative images of the invasion assay after scratches at 0 h, 10x magnification; the scale

including inflammation, cell migration, and invasion (Fig 1I, Tables S1 and S2). Notably, the hallmark pathway, EMT, was significantly enriched ($P < 0.05$), which coincides with the morphological phenotype we observed. We further investigated this hallmark pathway by examining the associated genes that were up-regulated and down-regulated in both UVB and poly(I:C) treatment (Fig 1J and K). Notable genes that were up-regulated in both treatment conditions included *CDH2*, *FN1*, *IL6*, many integrins, MMPs (*1*, *14*, *2*, and *3*), *NOTCH2*, *SNAI2*, *TGFB1*, *TIMP1*, *TIMP3*, and *VIM* (Fig 1J and K, Tables S3 and S4). In aggregate, TLR3 activation in primary human keratinocytes leads to an EMT-like morphological change and corresponding associated changes in gene transcription.

### Keratinocytes acquire an EMT-like signature after TLR3 activation

Next, we investigated whether our observed phenotype was indeed EMT. TLR3 induces the expression of key EMT-associated transcription factors: *ZEB1*, *SNAI1*, and *TWIST1* (Lamouille et al, 2014). Both UVB and poly(I:C) significantly increased the gene expression of all three transcription factors compared with the control ($P < 0.05$) (Fig 2A and B). We subsequently quantified changes in mesenchymal- and epithelial-associated proteins by immunoblot after UVB and poly(I:C) exposure. Fibronectin, a characteristic mesenchymal protein, significantly increased (~4-fold, $P < 0.05$), after both UVB and poly(I:C) exposure, in a time-dependent manner (Fig 2C and D). In addition, direct TLR3 activation via poly(I:C) significantly increased the expression of vimentin, an intermediate filament protein associated with EMT (Fig 2C and D). The epithelial marker, E-cadherin, remained stable in UVB-treated cells and decreased slightly 72 h after poly(I:C) treatment, suggesting a partial EMT in keratinocytes after TLR3 activation (Nieto et al, 2016). Together, these data suggest that the TLR3-induced "spindle-like" morphology in normal keratinocytes resembles EMT.

The ability of cells to migrate and invade is another feature of EMT (Lamouille et al, 2014). To determine whether TLR3 induced functional changes associated with the EMT-like phenotype in keratinocytes, we performed scratch assays to quantify migration and invasion in vitro. Keratinocytes were pre-treated with poly(I:C) for 24 h and then plated to form a confluent monolayer. The keratinocyte monolayer was scratched and monitored for wound closure (migration) using IncuCyte Live-Cell Analysis System (Fig 2E). The wound closure rate was slightly accelerated between 4 and 7 h post-scratch in poly(I:C)-treated keratinocytes compared with control keratinocytes (Fig 2F), suggesting that TLR3 activation eliminates the lag observed in normal wound closure. Furthermore, poly(I:C)-treated keratinocytes had a significantly increased capacity to invade compared with un-treated keratinocytes (Fig 2G and H). In sum, TLR3 activation minimally increased the migratory capacity and drastically

increased the invasive capability of normal keratinocytes. Taken together, our data support a TLR3-driven EMT in human keratinocytes.

### Activation of TLR4, TLR5, or TLR7 does not lead to an EMT-like phenotype

We next asked whether the EMT-like phenotype was specific to TLR3 activation or a general response to TLR activation. We investigated two cell surface TLRs expressed in human keratinocytes that recognize bacterial products (Lebre et al, 2007; Olaru & Jensen, 2010): TLR4, which recognizes LPS, and TLR5, which recognizes flagellin. TLR7, which recognizes single-stranded RNA (ssRNA), was also investigated using imiquimod, a TLR7 agonist used clinically to treat non-melanoma skin cancers (NMSCs) (Voiculescu et al, 2019). These TLR agonists did not induce the spindle-like morphological change in keratinocytes (Fig 3). TLR4 and TLR5 activation did not increase *VIM* or *ZEB-1* mRNA expression (Fig 3B and D), but TLR7 activation increased *VIM* and induced a transient increase in *ZEB-1* (Fig 3F), although the magnitude of increased gene expression was lower than poly(I:C) stimulation. These results suggest that the EMT-like morphology is not a universal feature of TLR activation in keratinocytes and is specific to TLR3 activation.

### TLR3 inhibition attenuates the expression of EMT-associated genes and proteins

As the EMT-like morphology and gene expression were specific to TLR3 activation, we determined whether blocking TLR3 attenuated the TLR3-induced EMT-like morphology in normal keratinocytes. We used both a pharmacological and a genetic approach to inhibit TLR3 activation and expression. CU CPT 4a, a competitive pharmacological inhibitor of TLR3, prevents increased *TLR3* gene and TLR3 protein expression seen with TLR3 activation in keratinocytes when applied 1 h before UVB or poly(I:C) (Fig S1A–C). The spindle-like morphology was also absent with this TLR3 inhibitor (Fig S1D and E). UVB-induced *VIM* and *ZEB1* gene expression and fibronectin and vimentin protein expression were also blocked or attenuated by the pharmacological inhibitor (Figs 4A and B and S1F). Similarly, the TLR3 inhibitor decreased poly(I:C)-induced EMT-associated mRNA and proteins compared with control cells (Figs 4C and D and S1G). Altogether, pharmacological inhibition of TLR3 mitigates the EMT-like morphology and prevents the TLR3-regulated gene expression of EMT genes.

To overcome possible off-target effects of the pharmacological inhibitor, we also generated primary human TLR3 knockdown (KD) keratinocytes using the CRISPR/Cas9 technology (Fig S2A). CRISPR/Cas9-generated KD keratinocytes were cultured with the Rho kinase inhibitor, Y-27632, allowing for prolonged lifespan and successful antibiotic selection of the primary CRISPR/Cas9 TLR3

bar represents 200 µm, n ≥ 5 technical replicate wells per group. **(H)** Graphical representation of relative wound density overtime for control and poly(I:C)-treated cells, n ≥ 19. Invasion data were analyzed by multiple t tests, one per time-point, using the Holm–Sidak correction method, with alpha = 0.05, and a consistent SD was assumed. * denotes significance compared with the control, $P < 0.05$.

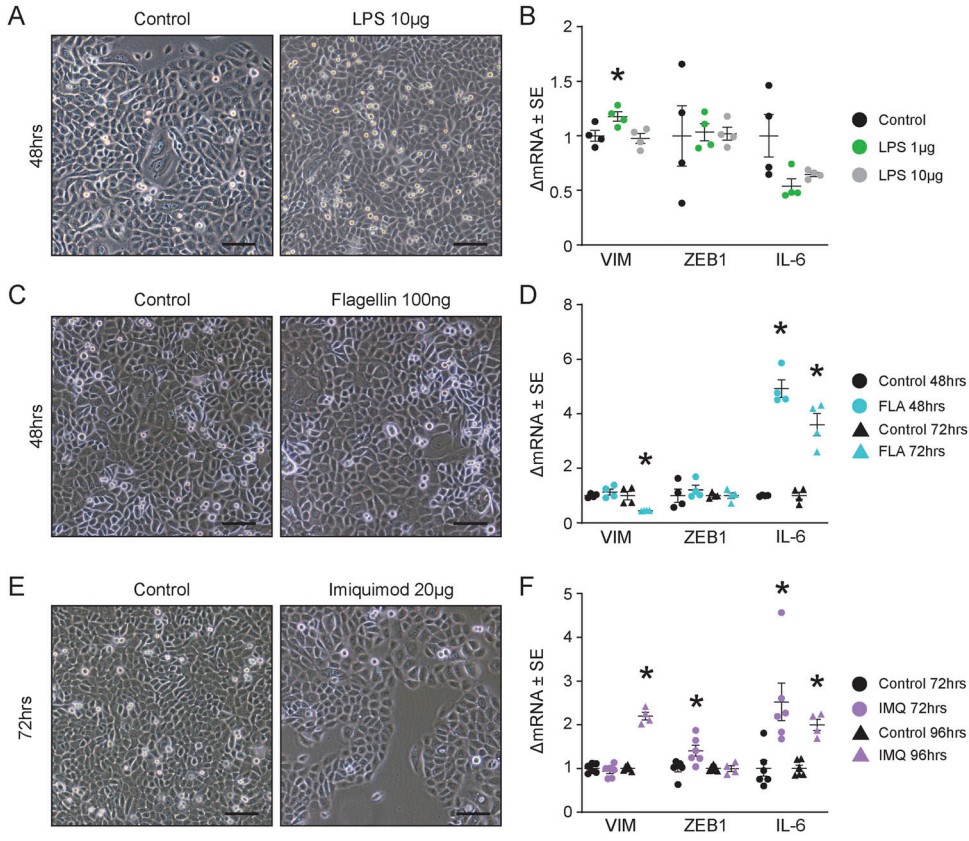

**Figure 3. Activation of TLR4, TLR5, or TLR7 does not lead to an epithelial-to-mesenchymal transition (EMT)–like phenotype.**
**(A)** Keratinocytes were treated for 1 h with LPS (1, 10 μg/ml), a TLR4 agonist, and underwent a 48-h washout treatment; representative images captured at 10X, scale bar = 100 μm. **(B)** qRT-PCR for EMT markers, *VIM* and *ZEB1*, and cytokine *IL6* for keratinocytes treated with LPS (1, 10 μg/ml) for 1 h followed by a 48-h washout, normalized to *RPLP0*, n = 4. **(C)** Keratinocytes treated for 24 h with flagellin (100 ng/ml), a TLR5 agonist, with a 24-h washout; representative images captured at 10X, scale bar = 100 μm. **(D)** qRT-PCR for EMT markers, *VIM* and *ZEB1*, and cytokine *IL6* for keratinocytes treated with flagellin (100 ng/ml) for 24 h followed by a 24-h or 48-h washout, respectively, normalized to *RPLP0*, n = 4. **(E)** Keratinocytes treated with imiquimod (20 μg/ml), a TLR7 agonist, for 24 h followed by a 48-h washout; representative images captured at 10X, scale bar = 100 μm. **(F)** qRT–PCR for EMT markers, *VIM* and *ZEB1*, and cytokine *IL6* for keratinocytes treated for 24 h with imiquimod (20 μg/ml) followed by a 48- or 72-h washout, normalized to *RPLP0*, n ≥ 4. * denotes significance compared with the control, $P < 0.05$, Mann–Whitney *t* test, two-tailed.

KD keratinocytes (Chapman et al, 2010; Nelson et al, 2016; Fenini et al, 2018; Grossi et al, 2020). We used three independent pooled populations of CRISPR TLR3 KD cells. All three pools of cells (B, C, and D) maintained detectable levels of full-length TLR3 protein expression, reduced by ~28% compared with non-CRISPR primary keratinocytes (Fig S2B). Importantly, these three independent pools of TLR KD keratinocytes lacked the substantial expression of active (cleaved) TLR3, reduced by ~93% when compared to non-CRISPR keratinocytes, even in the presence of poly(I:C) (Fig S2B). We used pool D keratinocytes for subsequent experiments because it showed the greatest reduction in both full-length and cleaved TLR3 protein expression (Fig S2B). Notably, CRISPR-KD of TLR3 prevented the EMT-like morphological changes induced by UVB or poly(I:C) (Fig S2C), similar to our findings with the pharmacological inhibitor. The decreased expression of active cleaved TLR3 in the CRISPR-KD cells is likely due to a disruption in the transport of the full-length TLR3 protein from the ER to the endolysosome, where proteolytic cleavage occurs, given that the CRISPR gDNA cut site on the TLR3 gene is upstream of an essential ubiquitination site needed for transport (Li et al, 2020). Furthermore, UVB-induced EMT gene and protein expression changes were also attenuated in TLR3 KD cells, but less so than in poly(I:C)-treated cells (Fig 4E and F). Similarly, TLR3 KD prevented poly(I:C)-induced increases in fibronectin and vimentin and decreases in E-cadherin protein expression (Fig 4G and H). However, we observed the increased numbers of dead cells in the UVB-treated CRISPR cells compared with non-CRISPR keratinocytes

potentially confounding these results; increased cellular stress from CRISPR methodology in conjunction with subsequent UVB exposure may contribute to this increase in cell death (Feehan & Shantz, 2016).

## Inflammatory cytokines downstream of TLR3 are not sufficient to induce EMT alone

Classically, TLR3 activates two downstream inflammatory signaling cascades: (1) NF-κB, which leads to the production of the inflammatory cytokines IL-6, IL-8, and TNF-α, and (2) interferon regulatory factor 3, leading to the production of type I and III IFNs, including IFN-α, IFN-β, and IFN-λ (Kawai & Akira 2010). Poly(I:C) significantly increased both IL-6 and IFN-λ in normal human epidermal keratinocytes (NHEKs) (Fig 5A and D). IL-6 and IFN-λ up-regulate a number of EMT-related genes in epithelial cancer models, suggesting that these two downstream activators may be involved in initiating the TLR3-mediated EMT in normal keratinocytes (Sullivan et al, 2009; Yadav et al, 2011; Mucha et al, 2014). However, direct stimulation with either IL-6 (10, 50, 100, and 200 ng/ml) or IFN-λ (50 and 100 ng/ml) showed no EMT-like morphological change (Fig 5B and E). EMT-associated gene expression increased significantly (<2-fold), although not as robustly as with poly(I:C)-induced gene expression (Fig 5C and F). These data suggest that TLR3-induced partial EMT requires more than IL-6 or IFN-λ alone to induce visible morphological changes.

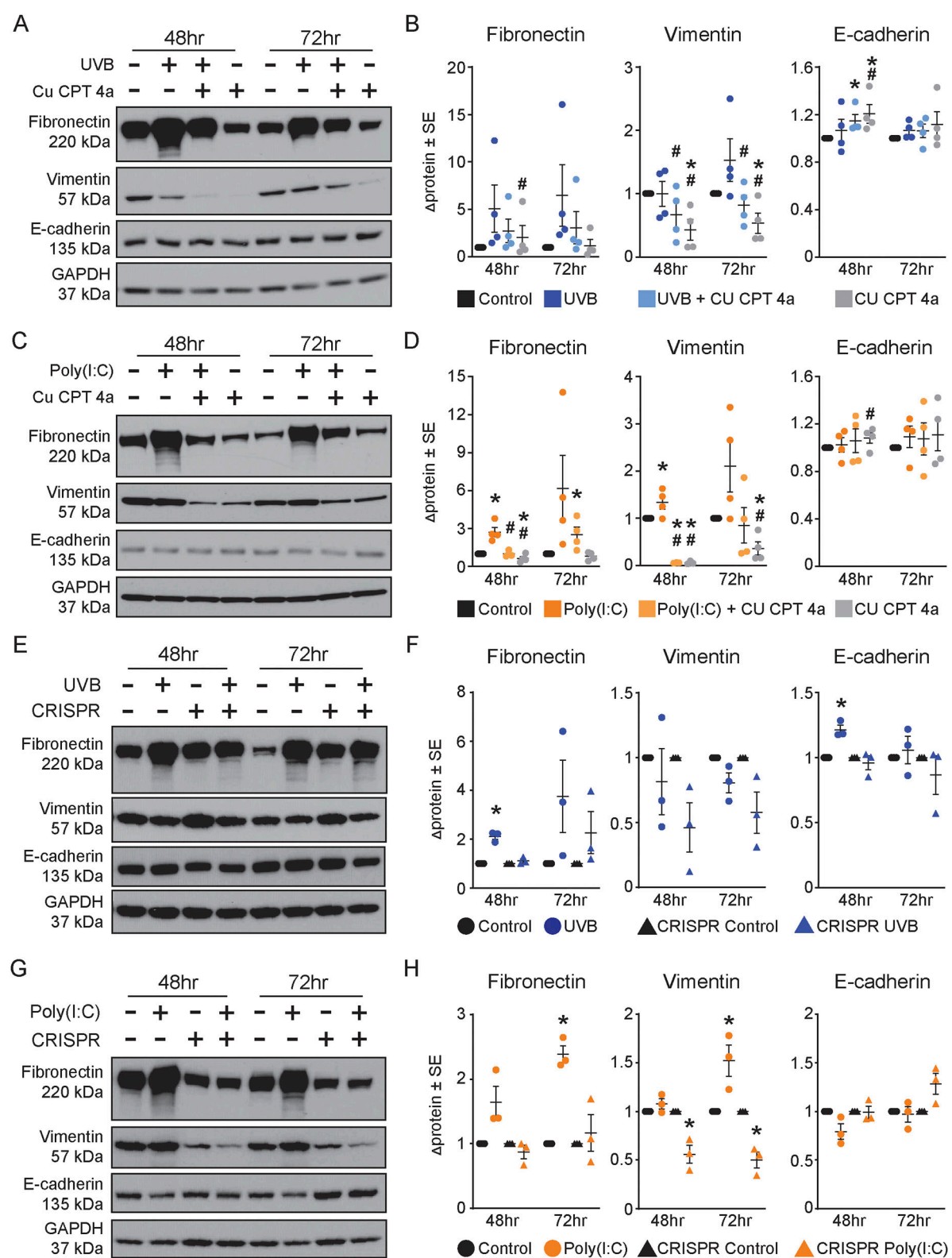

**Figure 4. Inhibition of TLR3 attenuates the expression of epithelial-to-mesenchymal transition (EMT)–associated factors.**
**(A)** Treatment with chemical inhibitors limits EMT-associated protein changes induced by UVB treatment at 48 and 72 h, representative image. **(B)** Protein level was semi-quantified by densitometric analysis. Samples were normalized to GAPDH; fold change was calculated compared with the control, n ≥ 3. **(C)** Treatment with chemical inhibitors limits EMT-associated protein changes induced by poly(I:C) treatment at 48 and 72 h, representative image. **(D)** Protein level was semi-quantified by densitometric analysis. Samples were normalized to GAPDH; fold change was calculated compared with the control, n ≥ 3. **(E)** TLR3 KD CRISPR keratinocytes have

### NF-κB signaling is required for EMT after TLR3 activation

We observed a number of significant changes in genes downstream of NF-κB in cells treated with either UVB or poly(I:C) (Fig 6A and B, Tables S5 and S6). UVB is known to activate TLR3 signaling and NF-κB in keratinocytes (Adhami et al, 2003; Lewis & Spandau, 2007; Bernard et al, 2012). This observation, coupled with published evidence that NF-κB increases EMT-associated genes (Lilienbaum & Paulin, 1993; Huber et al, 2004; Min et al, 2008; Tian et al, 2017), led us to test the hypothesis that NF-κB was involved in the poly(I:C)-induced TLR3-mediated EMT phenotype. Keratinocytes treated with poly(I:C) significantly increased the nuclear localization of NF-κB (p65) compared with the control cells (39.5% vs 6.9%, P < 0.05), consistent with previous studies (Lebre et al, 2007). Pre-treatment with SC-514, a pharmacological inhibitor of IKK2 and N-κB signaling, before poly(I:C) suppressed NF-κB p65 nuclear translocation to that of control levels (5.4%) (Figs 6C and D and S3). SC-514 also reduced the expression of fibronectin and vimentin compared with poly(I:C)-treated keratinocytes (Fig 6E and F) and blocked invasion in the in vitro invasion assay (Fig 6G). These data strongly suggest that NF-κB activation mediates the EMT response after TLR3 activation in keratinocytes.

### Human basal cell carcinomas (BCCs) and SCCs exhibit unique TLR3 protein expression patterns

As UVB up-regulates TLR3 expression in murine skin (Bernard et al, 2012), we determined whether TLR3 expression was increased in clinically normal, sun-exposed skin in humans. TLR3 gene expression was increased in sun-exposed skin (face) compared with non–sun-exposed skin (breast) (Fig 7A). Prior studies have shown that TLR3 activation in keratinocytes increases Wnt and SHH pathways, and TLR3 activation in mesenchymal stem cells increases Notch signaling pathways associated with NMSCs (Nelson et al, 2015; Rashedi et al, 2017). Therefore, we determined whether the characteristic NMSC-associated gene expression of *GLI1*, *NOTCH1*, and *SHH* was also increased in sun-exposed skin compared with non–sun-exposed skin (Fig 7A). Both TLR3- and NMSC-associated genes were increased in sun-exposed skin, setting up the possibility that TLR3 activation may contribute to NMSC-associated gene expression. We next tested the hypothesis that TLR3 activation independent of UVB exposure is sufficient to alter NMSC-associated gene expression in keratinocytes. The gene expression of *GLI1* and *SHH* was significantly increased with TLR3 activation, and conversely, *NOTCH1* gene expression was significantly decreased in treated keratinocytes compared with the control (Fig 7B). TLR3 inhibition with CU CPT 4a blocked poly(I:C)-induced increases in *TLR3*, *GLI1*, *SHH*, and *NOTCH* gene expression (Fig 7C). This gene expression profile suggests that direct TLR3 activation can lead to changes in

NMSC-associated genes, independent of additional stimuli incurred by UVB exposure. Together, these data show increased TLR3 expression in clinically normal, sun-exposed skin and TLR3 activation in cultured keratinocytes leads to NMSC-associated gene expression.

TLR3 is expressed in normal skin, and TLR3 mRNA expression is increased in NMSCs compared with normal skin (Muehleisen et al, 2012). Using immunohistochemistry, we stained 21 established cutaneous squamous cell carcinomas (cSCCs) and BCCs for TLR3 protein to confirm its expression and localization within tumors. Mohs micrographic surgery tumor samples were collected by standard of care under an approved IRB protocol at Penn State College of Medicine. Tumor type and histological subtype were confirmed with hematoxylin and eosin (H&E) staining by a board-certified dermatopathologist. TLR3 staining patterns (tumor edge or interior) and intensity (0—none; 1—faint; 2—moderate; and 3—intense staining) in each tumor sample were scored by two blinded investigators (Table S7). Well-differentiated cSCCs had stronger TLR3 staining at the tumor edge than within the interior of the tumor, whereas moderately differentiated cSCCs had consistent TLR3 staining throughout the entire tumor (Fig 7D). BCCs, regardless of histological subtype, had strong, uniform TLR3 protein expression throughout the tumor (Fig 7E). BCCs had more intense, uniform TLR3 staining than cSCCs, which were more variable in TLR3 expression and location (Table S7). Similar to prior studies, we note p65 NF-κB nuclear localization in BCC and SCC tumors, but the expression was highly variable within a tumor and between tumors (Dajee et al, 2003; Weng et al, 2013; Li et al, 2022; Jenni et al, 2023). Unlike other epithelial cancers, we found no correlation between TLR3 staining intensity or location and histological subtype in either cSCCs or BCCs. Overall, BCCs and cSCCs exhibit unique TLR3 staining patterns and intensities that are specific to their tumor types.

## Discussion

In this study, we demonstrated that TLR3 activation in normal keratinocytes provokes a morphological change with accompanying EMT hallmarks: increased expression of key EMT-associated genes and increased migration and invasion capabilities. These TLR3-induced EMT features are mediated, in part, by NF-κB signaling. These data highlight that in addition to classical activation of innate immune signaling, TLR3 activation in normal keratinocytes activates EMT programming, not only a normal developmental process, but also a process involved in cell transformation and cancer. This sets up the possibility that TLR3 signaling is both helpful, in terms of innate immunity and protection against pathogens, and potentially harmful if developmental programs are activated out of context. These findings provide a mechanistic link between TLR3 signaling, UVB-induced skin damage, and

decreased EMT-associated protein changes induced by UVB treatment at 48 and 72 h compared with normal controls, representative image. **(F)** Relative density of fibronectin, vimentin, and E-cadherin was normalized to GAPDH. Fold change was calculated compared with the control, n = 3. **(G)** TLR3 KD CRISPR keratinocytes have decreased EMT-associated protein changes induced by poly(I:C) at 48 and 72 h compared with normal controls, representative image. **(H)** Relative density of fibronectin, vimentin, and E-cadherin was normalized to GAPDH. Fold change was calculated compared with the control, n = 3. * denotes significance compared with the control, # significance compared with UVB or poly(I:C), paired *t* test, one-tailed, *P* < 0.05.

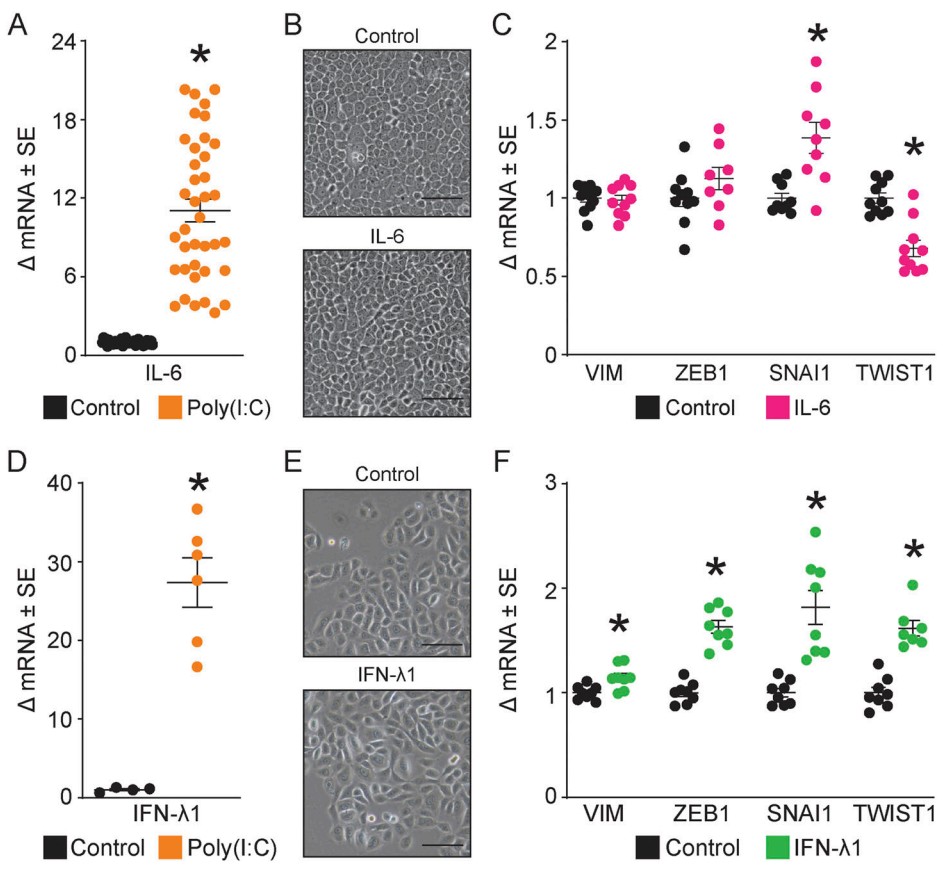

**Figure 5. IL-6 and IFN-λ1 are not sufficient for epithelial-to-mesenchymal transition (EMT)–like change.**
**(A)** *IL6* (16-fold) mRNA compared with controls by qRT–PCR after poly(I:C) treatment, 24 h of treatment with a 48-h washout, normalized to *RPLP0*, n ≥ 30. **(B)** Cell morphology after rhIL-6 (50 ng/ml) treatment for 72 h. Representative images were captured at 10x magnification; the scale bar represents 100 μm. Additional IL-6 concentrations (10, 50, 100, and 200 ng/ml) tested did not alter cell morphology. **(C)** qRT–PCR of EMT-associated genes (*VIM*, *ZEB1*, *SNAI1*, *TWIST1*) after recombinant (rh) rhIL-6 (50 ng/ml) treatment. Data were normalized to *RPLP0*, Mann–Whitney *t* test, one-tailed, *P* < 0.05, n ≥ 8. **(D)** *IFNL1* (27-fold) mRNA compared with controls by qRT–PCR after poly(I:C) treatment, 24 h of treatment with a 48-h washout, normalized to *RPLP0*, n ≥ 4. **(E)** Cell morphology after rhIFN-λ1 (50 ng/ml) treatment for 72 h. Representative images were captured at 10x magnification; the scale bar represents 100 μm. Additional IFN-λ1 concentrations (50 and 100 ng/ml) tested did not alter cell morphology. **(F)** qRT–PCR of EMT-associated genes (*VIM*, *ZEB1*, *SNAI1*, *TWIST1*) after recombinant rhIFN-λ1 (50 ng/ml) treatment. Data were normalized to *RPLP0*, Mann–Whitney *t* test, one-tailed, *P* < 0.05, n ≥ 6. * denotes significance compared with the control, *P* < 0.05, Mann–Whitney *t* test, two-tailed, unless otherwise noted above.

keratinocyte morphology. It is plausible that these TLR3-induced keratinocyte morphology and gene expression programs contribute to keratinocyte transformation and subsequent NMSC development.

Published studies show a clear dichotomy for the role of TLR3 in epithelial malignancies, but its impact on normal keratinocyte physiology or within NMSC is largely unexplored (Muresan et al, 2020). We investigated the mechanism behind TLR3-activated EMT in normal keratinocytes and observed significant activation of the NF-κB pathway (Fig 6). NF-κB increases EMT-associated genes *SNAI1*, *TWIST1*, *ZEB1*, and *VIM* in a variety of disease processes including pulmonary lung fibrosis and many cancers: head and neck, prostate, pancreatic, and breast cancer (Lilienbaum & Paulin, 1993; Huber et al, 2004; Min et al, 2008; Tian et al, 2017). Although the exact contribution of NF-κB signaling in cSCCs is unclear (Dajee et al, 2003; Zhang et al, 2004; Kim & Pasparakis, 2014; Watt et al, 2015), our data support the idea that TLR3-mediated NF-κB activation in normal keratinocytes promotes EMT, as blocking NF-κB activity attenuates the expression of EMT genes/proteins and invasion properties. Controlling acute inflammation, of which NF-κB activity is a major contributor, after sun exposure may be important for blocking the early events in NMSC formation. Preventing TLR3 recognition of dsRNA from damaged keratinocytes could be an important step in limiting NF-κB activation and NMSC development.

With the increase in the expression of TLR3 and genes characteristic of NMSC in clinically normal human sun-exposed skin

compared with non–sun-exposed skin, we hypothesized that TLR3 activation in keratinocytes could be an important initiating event, which may prime them for the next "hit" and subsequent tumor development. As such, we focused on how TLR3 activation and signaling affect normal primary keratinocytes. It was important for our studies to understand the impact of TLR3 activation in keratinocytes that did not have any initiated properties, like the more commonly used keratinocyte cell lines with known TP53 mutations and altered NF-κB signaling (Lehman et al, 1993; Lewis et al, 2006; Takada et al, 2017). As predicted, independent donors of keratinocytes had variable TLR3 expression and response to UVB and poly(I:C) exposure; this variability in primary cells is documented in the mRNA expression of TLRs between patients (Olaru & Jensen, 2010). Significant evidence supports the interpersonal variability of the TLR3 gene between individuals with different SNPs correlating with an increased risk of certain diseases and the function of TLR3 (Qi et al, 2010). When placed in the context of our findings, variability in TLR3 function may be a contributing factor in NMSC development, and influence an individual's response to UVB exposure. Understanding how variations in TLR3 expression and activation impact NMSC development could provide a personalized approach to identifying groups that are at high risk of NMSC development.

In breast, prostate, and esophageal carcinomas, the expression of TLR3 correlates with tumor aggressiveness and poor clinical outcomes related to TLR3's ability to promote cancer stem cell survival, cell invasion, and inflammation (Sato et al, 2009; Jia et al,

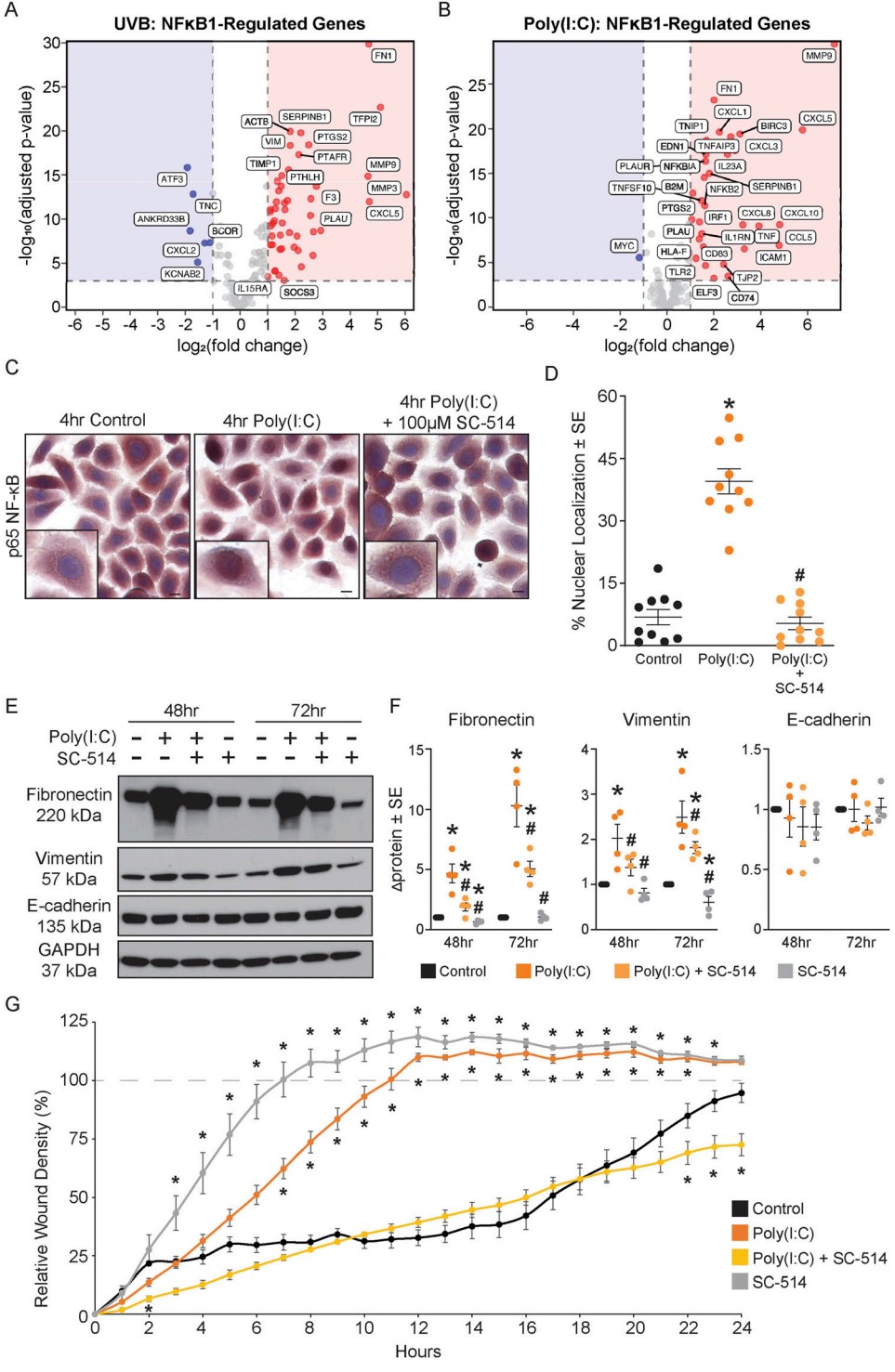

**Figure 6. NF-κB signaling is required for epithelial-to-mesenchymal transition after TLR3 activation.**

**(A, B)** Volcano plots illustrating changes in NF-κB-1–regulated genes after (A) UVB and (B) poly(I:C) treatment. **(C)** Normal human epidermal keratinocytes treated with 100 μM SC-514 (IKK2 inhibitor) for 1 h before 4 h of poly(I:C) treatment (20 μg/ml) were stained for NF-κB(p65) localization. Representative images were captured at 10X, insets to highlight + or − nuclear location of p65. Scale bars represent 10 μm. **(D)** Graphical representation of image quantification (n = 10 per group), unpaired t test, two-tailed. * denotes significance compared with the control, P < 0.05, # significance compared with poly(I:C), P < 0.05. **(E)** Treatment with SC-514 IKK2 (100 μM) limits epithelial-to-mesenchymal transition–associated protein changes induced by poly(I:C) treatment at 48 h and 72 h, representative image. **(F)** Relative density of fibronectin, vimentin, and E-cadherin was normalized to GAPDH. Fold change was calculated compared with the control, paired t test, one-tailed, n = 4. **(G)** Representative invasion assay with normal human epidermal keratinocytes treated with SC-514, n ≥ 5 technical replicate wells per group. Data were analyzed by multiple t tests, one per time-point, using the Holm–Sidak correction method, with alpha = 0.05, and a consistent SD was assumed. * denotes significance compared with the control, # significance compared with poly(I:C), P < 0.05.

2015; Bugge et al, 2017). In contrast, in melanoma and head and neck SCCs, TLR3 activation is associated with good prognosis and tumor regression, likely because of TLR3 activation of apoptosis-inducing mechanisms (Salaun et al, 2007; Nomi et al, 2010). TLR3 and *TLR3* mRNA expression is up-regulated in NMSCs, suggesting that TLR3 signaling may play a role in NMSC initiation, formation, or development (González-Reyes et al, 2010, 2011; Sheyhidin et al, 2011; Muehleisen et al, 2012). Here, we confirmed that TLR3 protein expression is elevated and sustained in NMSCs compared with the surrounding stroma. In addition, we noted variability in TLR3 expression and location in our set of NMSCs. Although BCCs and cSCCs had varying degrees of TLR3 expression, there were also differences

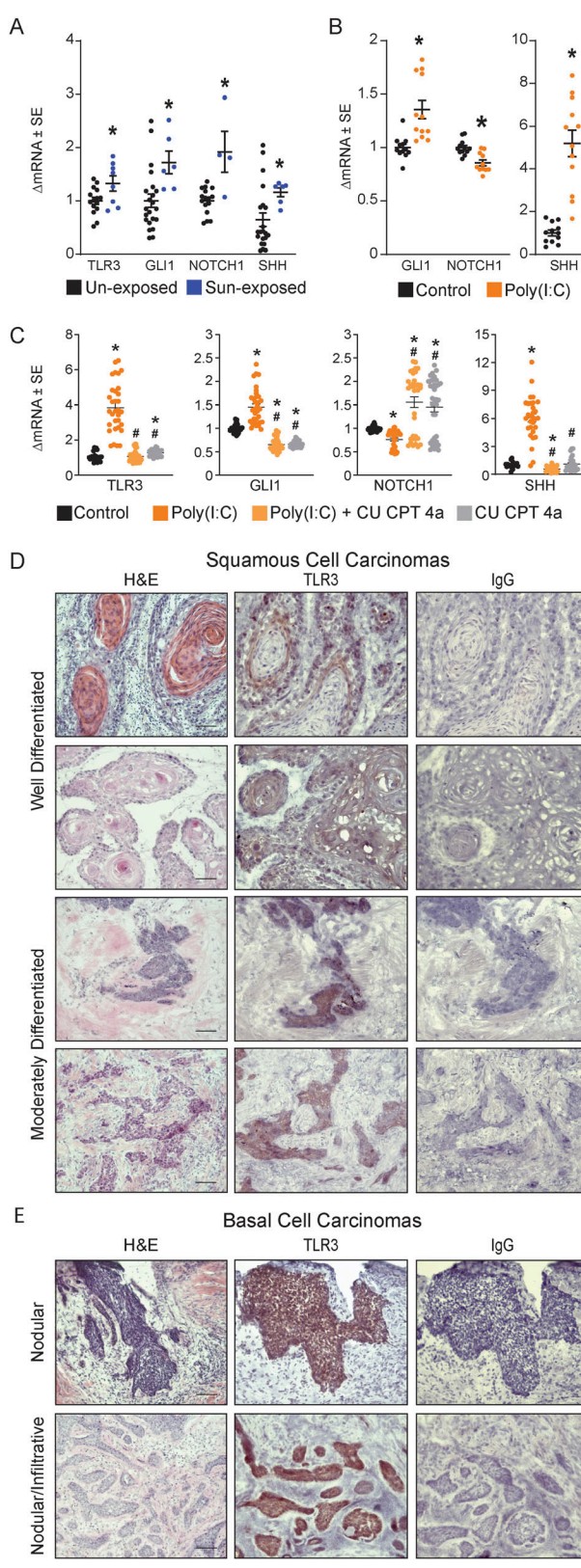

**Figure 7. Patient tumors express TLR3.**
(A) Gene expression of *TLR3* and non-melanoma skin cancer (NMSC)–associated genes in clinically normal skin from both unexposed and sun-exposed body areas, qRT–PCR, normalized to *RPLP0*, n ≥ 4 patient samples per

in the location of the TLR3-positive cells. The expression of TLR3 in NMSCs certainly suggests that the response we observed in primary keratinocytes may be an early step in keratinocyte transformation. However, we did not investigate the role of TLR3 in established skin cancers or how TLR3 may act in the fully transformed keratinocytes of BCCs and SCCs. We hypothesize that TLR3 protein expression, and ultimately activity, in BCCs and SCCs may be different than in healthy keratinocytes. Factors that can influence this expression and activity are likely the tumor microenvironment, or the origin of the keratinocyte population in BCCs and SCCs. Further investigation into TLR3 staining patterns in patient tumors could provide new approaches to understanding tumor development and aggressiveness. We did note increases in TLR3 expression in healthy sun-exposed skin compared with more sun-naïve areas, suggesting that TLR3 activation from sun exposure is a long-term process. Actinic keratoses (AKs) are thought to be a precursor lesion to cSCCs with data suggesting that ~0.5% transition to cSCCs (Green & Olsen, 2017). Further studies investigating the TLR3 expression of these AK lesions are necessary, and it is plausible that high levels of TLR3 expression and activity in AKs may be a predictor of which lesions develop into cSCCs.

In conclusion, we demonstrate that TLR3 activation in normal healthy keratinocytes leads to a partial EMT response with enhanced migration and invasion properties. These gene and morphological changes in response to TLR3 activation in normal keratinocytes may prime keratinocytes for subsequent development into transformed keratinocytes or NMSCs (with additional insults/factors) if not quickly removed by either the skin-intrinsic or immune-dependent protection mechanisms of apoptosis (Rangwala & Tsai, 2011; Feehan & Shantz, 2016). Moreover, this work supports a broader role of TLR3 in cell developmental programming outside of its more classically known role in innate immune response activation.

## Materials and Methods

### Cell culture

NHEKs were obtained from fresh foreskins (IRB #118) using standard isolation protocols (Aasen & Izpisúa Belmonte, 2010; Nelson et al, 2015) or from commercially available pooled neonatal human

group. (B) Gene expression of NMSC-associated genes in normal human epidermal keratinocytes treated with TLR3 agonist poly(I:C) (20 μg/ml) for 24 h followed by a 24-h washout. Data were normalized to *RPLP0*, n ≥ 11. (C) Gene expression of *TLR3*- and NMSC-associated genes in normal human epidermal keratinocytes pre-treated with CU CPT 4A (80 μM) for 1 h before poly(I:C) (20 μg/ml) treatment. Cells were harvested 48 h after poly(I:C). Data were normalized to *RPLP0*, # significance compared with poly(I:C), *P* < 0.05, n ≥ 20. (D) TLR3 protein expression in well- and moderately differentiated cutaneous squamous cell carcinomas. Tumors were graded by H&E and stained for TLR3 and IgG by IHC (n = 13 individual tumors), images were captured at 20X, and the scale bar represents 50 μm. (E) TLR3 protein expression in graded basal cell carcinomas. Tumors were graded by H&E and stained for TLR3 and IgG by IHC (n = 8 individual tumors), images were captured at 20X magnification, and the scale bar represents 50 μm. * denotes significance compared with the control, *P* < 0.05, Mann–Whitney *t* test, two-tailed.

epidermal keratinocytes (Lonza). Each experiment was conducted with pooled keratinocytes from at least three independent donors in duplicate/triplicate replicates (Supplemental Data 1). Keratinocytes were cultured in KGM-GOLD culture medium with all supplements (Lonza). Keratinocytes were seeded at an equal density of 25,000 cells per well in a six-well plate and treated, whether for control or treatment groups, at 30% confluence. Treatment with poly(I:C) 20 µg/ml (InvivoGen) was applied in basal medium containing transferrin, hydrocortisone, and antibiotic (KGM+H, T, GA) for up to 24 h; after that, media were replaced with KGM-GOLD as described previously (Nelson et al, 2015). Control groups were also transitioned to KGM+H, T, GA plus sterile water (in a volume to match that of the poly(I:C)) for the same duration as the treatment group. Total RNA and protein were isolated at time-points indicated in each figure.

## Reagents

Poly(I:C) HMW (high molecular weight) (synthetic analog of dsRNA—TLR3 ligand) was purchased from InvivoGen, and a stock solution of 1 mg/ml was made in sterile water according to the manufacturer's instructions and stored at −20C until use. TLR3 pharmacological inhibitor (CU CPT 4a; (R,Z)-2-(((3-chloro-6-fluorobenzo[b]thiophen-2-yl)(hydroxy)methylene) amino)-3-phenylpropanoic acid) was obtained from Tocris Biologicals, and a stock solution of 100 mM was made in DMSO and stored at −20C. Y-27632 (Rho kinase inhibitor) was purchased from Sigma-Aldrich, and a stock solution of 10 mM was made in sterile water and stored at 4C until use. IKK2 inhibitor, SC-514 (Calbiochem), was purchased from Sigma-Aldrich, and a stock solution of 50 mM was made in sterile DMSO according to the manufacturer's instructions and stored at −20C. LPS from *Escherichia coli* O127:B8 (LPS), TLR4 agonist, was purchased from Sigma-Aldrich, and a stock solution of 5 mg/ml was made in sterile water and stored at 4C. Flagellin from *P. aeruginosa*—TLR5 ligand—was purchased from InvivoGen, and a stock solution of 100 µg/ml was made in sterile water and stored at −20C until use. Imiquimod (R837, small synthetic antiviral molecule—TLR7 ligand) was purchased from Invivogen, and a stock solution of 1 mg/ml was made in sterile water and stored at −20C until use. Recombinant human IL-6 (R&D) was made in sterile PBS at 100 µg/ml and stored at −20C. Recombinant human IL-29/IFN-*λ*1 (R&D) was made in sterile PBS at 100 µg/ml and stored at −20C.

## UVB

NHEKs were exposed to UVB (FS20 UVB bulbs; National Biological) at a dose of 10 mJ/cm$^2$. A UVB 500C meter (National Biological) was used to measure bulb intensity and calculate the time for dose before each experiment. Immediately after exposure, fresh KGM-GOLD media were added back to the keratinocytes. For cells pretreated with TLR3 inhibitor (CU CPT 4a), the media containing the inhibitor were added back to the cells immediately after UVB exposure. After 24 h, all cells received fresh KGM-GOLD media with no treatments.

## Generation of CRISPR/Cas9 NHEKs

The all-in-one pLentiCRISPR v2 plasmid containing sgRNA for TLR3 was purchased (gRNA sequence, TTCAACGACTGATGCTCCGA; GenScript), and viral production was performed using ViraPower

Lentiviral Expression Systems (Life Technologies) according to the manufacturer's protocols. Plasmids were cotransfected into HEK293T cells to generate virus. Independent pools of NHEKs were incubated with the virus in KGM-GOLD media with 10 µM Y-27632 and a 10 µg/ml polybrene (stock conc. 10 mg/ml, hexadimethrine bromide; Sigma-Aldrich). The medium was changed 48 h after transduction to keratinocyte media (Chapman et al, 2010) and selected using 1 µg/ml puromycin (3 mg/ml stock; Sigma-Aldrich) for 72 h in culture with 10 µM of inhibitor Y-27632. After selection, keratinocytes were passaged in keratinocyte media with 10 µM Y-27632 to generate three independent TLR3 CRISPR pools (B, C, and D). Protein validation of TLR3 knockdown is provided in Fig S2. This protocol was adapted from Fenini et al (2018) and Grossi et al (2020). Immediately before experimentation, cells were grown in KGM-GOLD without inhibitor Y-27632 and treatment with poly(I:C) and UVB was done as described above.

## qRT–PCR

RNeasy Mini Kit (QIAGEN) with DNase I digestion was used for total RNA isolation from NHEKs. RNA was reverse-transcribed to cDNA using the High Capacity Reverse Transcription kit (Applied Biosystems; Life Technologies). Gene expression was determined by qRT–PCR using TaqMan reagents for genes of interest (*GLI1* Hs01110766_m1, *NOTCH1* Hs01062014_m1, *TLR3* Hs01551078_m1, *IL-6* Hs00985639_m1, *VIM* Hs00958111_m1, *ZEB1* Hs00232783_m1, *SNAI1* Hs00195591_m1, *TWIST1* Hs00361186_m1, *IFNL1* Hs00601677_g1) and housekeeping gene ribosomal protein, large P0 (*RPLP0* Control Mix; TaqMan). Differences were assessed by comparative $\Delta\Delta C_T$ values with fold change calculations.

## Immunoblotting and ELISA

Cell lysates were collected in RIPA buffer (Sigma-Aldrich) with 7X protease inhibitor (Roche) and flash-frozen in liquid $N_2$. Lysates were broken up mechanically with a needle and syringe. Protein concentration was determined by Pierce BCA Protein Assay Kit (Thermo Fisher Scientific). Equal amounts of protein were loaded onto a 4–12% gradient Bis-Tris gel (NuPAGE). Proteins were transferred to a PVDF membrane and probed. Primary antibodies used were as follows: TLR3 (D10F10; CST), vimentin (D21H3; CST), fibronectin (ab45688; Abcam), E-cadherin (24E10; CST), and GAPDH (14C10; CST). An ELISA (R&D Systems) was used according to the manufacturer's directions to assay IL-6 protein levels in the media of cells exposed to UVB or poly(I:C).

## Migration and invasion assays

Wound healing assays were performed using IncuCyte Live-Cell Analysis System (Sartorius), and methods were adapted from the manufacturer's protocol and Zhou et al (2020). Cells were treated with poly(I:C) (20 µg/ml) for 18 h in KGM+H, T, GA media before plating in IncuCyte ImageLock 96-well plates (4379; Sartorius) at a concentration of 70,000 cells/well. Wells were coated with a

coating matrix (Gibco) for migration assays or 1:60 dilution of Matrigel for invasion assays. Cells were treated with poly(I:C) in KGM+H, T, GA media for 6 h before the scratch in the 96-well plate for a total treatment of 24 h. Scratches were made in each well using IncuCyte 96-well WoundMaker Tool (4563; Sartorius). For migration assays, wells were washed once in PBS and 100 $\mu l$ KGM-Gold media were added to each well. For invasion assays, wells were scratched as described and washed once in PBS, and 50 $\mu l$ Matrigel (1:10 dilution in KGM-Gold media) was added to each well. Plates were incubated at 37°C for 30 min to allow the Matrigel to gel, and then, 50 $\mu l$ media were added to each well. Images were taken every hour for 12–48 h, and relative wound density was quantified using IncuCyte Cell Migration Analysis software (9600-0012; Sartorius).

### RNA sequencing

Total RNA was isolated from cells using RNeasy Mini Kit (QIAGEN) with DNase I digestion. RNA samples were sent to the Genomics Core Facility (Penn State, University Park) for quality testing by Bioanalyzer (all samples had RNA integrity number [RIN] 10), library preparation, and sequencing. A barcoded library was created with inserts on the 3′ end with the Lexogen QuantSeq kit. RNA sequencing was performed on Illumina HiSeq, with a single read, rapid run, and 100-nucleotide read length. A total of 119.7 million reads were generated from 12 total RNA samples (72 h UVB [10 mJ/cm$^2$ and control] and 72 h poly(I:C) [20 $\mu g$/ml and control], n = 3 per group). Sequencing quality was confirmed using FastQC and

downloaded from Ensembl and were used to generate an indexed genome for alignment using STAR (v2.7.9a) (Dobin et al, 2013; Cunningham et al, 2022). After alignment, reads were quantified using featureCounts and were exported as a table for subsequent analysis (Liao et al, 2014).

For differential expression analysis, the edgeR package was used to filter out lowly expressed genes and to calculate per-sample library size normalization factors (Robinson et al, 2010). Limma was then used to transform (voom) the count data and to test for gene-wise differential expression (Law et al, 2014; Ritchie et al, 2015). Differentially expressed genes were defined as having a Benjamini–Hochberg adjusted $P$-value < 0.05. For gene set enrichment analysis (GSEA), all genes were ranked by the moderated t-statistic returned by the Limma eBayes function. GSEA was performed using the fgsea package (v1.20.0) with gene set collections downloaded from the Molecular Signatures Database (MSigDB) using the msigdbr package (v7.5.1) (Subramanian et al, 2005; Liberzon et al, 2015; Korotkevich et al, 2021 Preprint; Dolgalev, 2022). Separately, pathway and transcription factor (TF) activities were estimated from the ranked gene lists using the PROGENy and DOROTHEA packages, respectively (Schubert et al, 2018; Garcia-Alonso et al, 2019).

Quality control, trimming, alignment, and read quantification steps were implemented in a Snakemake (v6.15.3) workflow (Mölder et al, 2021) and were executed using the Penn State College of Medicine High Performance Computing (HPC) system. Differential expression, GSEA, pathway, and TF analyses were performed in RStudio (2022.02.0+443 "Prairie Trillium" Release for macOS, R version 4.1.3).

**Table 1. RNA-sequencing reads and % alignment for each sample.**

| Group | Condition | Replicate | Total sequences (million) | % aligned | M aligned (million) |
|---|---|---|---|---|---|
| 72-h UVB | Control | A | 9.2 | 86.10% | 8 |
| | | B | 9.4 | 86.50% | 8.2 |
| | | C | 11.8 | 84.40% | 10 |
| | 10 mJ/cm$^2$ | A | 7.8 | 82.40% | 6.4 |
| | | B | 9.3 | 85.60% | 8 |
| | | C | 10.8 | 85.70% | 9.3 |
| 72-h poly(I:C) | Control | A | 11.4 | 84.80% | 9.7 |
| | | B | 8.9 | 86.80% | 7.7 |
| | | C | 11.8 | 87.00% | 10.2 |
| | 20 µg/ml | A | 9.5 | 87.50% | 8.3 |
| | | B | 12.2 | 88.20% | 10.8 |
| | | C | 7.6 | 86.80% | 6.6 |
| Average | | | 9.98 | 85.98% | 8.60 |

MultiQC (Andrews, 2010; Ewels et al, 2016). Trim Galore was used to remove low-quality bases from the 3′ end of the sequence (threshold Phred score = 10) and to remove Illumina adapter contamination (Martin, 2011; Krueger, 2021). Human reference genome (GRCh38.p13) and gene annotation (v105) files were

### Immunohistochemistry, immunocytochemistry, and histology

De-identified NMSC tumor samples were obtained under an IRB protocol (Study #118). Tumors were frozen in OCT and sectioned for staining. Immunohistochemistry (IHC) was performed on

frozen tumor sections using peroxidase detection with the ImmPACT NovaRED HRP substrate kit (Vector Laboratories). Antibodies were applied overnight: TLR3 (ab62566; Abcam) and rabbit IgG (ab172730; Abcam). Sections were counterstained with hematoxylin. Standard H&E staining was performed, and histology and tumor grade were assessed by a dermatopathologist. Images were captured at 20x magnification using Zeiss AxioCam ICc5 with Zen software on an Olympus BX40 microscope. TLR3 staining patterns (edge, interior, both) and intensity of staining (0—none; 1—faint; 2—moderate; and 3—intense staining) were graded by two reviewers who were blinded to the tumor type and stage (Table S1).

Immunocytochemistry was performed on NHEKs plated in 12-well culture plates. Cells were fixed in 4% paraformaldehyde after treatment with SC-514 and poly(I:C) as described in the legend. Fixed cells were incubated in primary antibodies overnight: p65 NF-κB (D14E12; CST) and isotype rabbit IgG (ab172730; abcam). Peroxidase detection as described above was used. Images were captured at 20x for quantification and 40x for display on a Nikon Eclipse Ti-S scope using NIS-Elements D 3.2 software. Nuclear localization of p65 was assessed by eye.

### Study approval

De-identified NMSC tumor samples were obtained under an IRB protocol approved by the Penn State College of Medicine Institutional Review Board (Study #118). Research was designated "Not Human Research," and no written informed consent was required.

### Statistics

Each experiment was conducted with pooled keratinocytes from at least three independent donors in duplicate or triplicate technical replicates. Complete data on the number of biological and technical replicates for each experiment are included in Supplemental Data 1. Biological replicates represent individual experiments that consisted of three or more pooled foreskins. Technical replicates were individual cell wells treated at the same time. For qRT–PCR experiments, a second group of technical replicates was incorporated at the level of plating cDNA. In the table, technical replicates include the multiple wells used during each experiment. A greater than or equal to sign represents when individual data or samples have been removed using our statistical outlier test.

Data were analyzed using GraphPad Prism software version 9. Data were checked for statistical outliers using the Rout method (Q = 1%). Outliers identified were removed, and datasets were checked for normal distribution using the Shapiro–Wilk normality test. For qRT-PCR, the data were analyzed by non-parametric Mann–Whitney $t$ tests. For normally distributed Western blot data, paired $t$ tests were used. All statistical tests are noted in the legend. A two-tailed test was used except in instances where we had a directional hypothesis. In the case of EMT changes, we used a one-tailed test where we expected mesenchymal markers (*ZEB1*, *SNAI1*, *TWIST1*, vimentin, fibronectin) to increase and epithelial markers to decrease (E-cadherin). Statistical significance was considered at $P < 0.05$. Migration and invasion assay data were analyzed by multiple $t$ tests, one per time-point, using the Holm–Sidak correction method, with alpha = 0.05, and a consistent scatter was assumed.

## Data Availability

RNA-sequencing data are deposited at Sequence Read Archive (https://www.ncbi.nlm.nih.gov/sra) under the accession number: PRJNA1086900.

## Supplementary Information

## Acknowledgements

We thank Kim Rodkey, BS, from our Dermatology Mohs team for tumor sample acquisition and preparation. The authors thank Andrew Kowalczyk, Ph.D., Sara Stahley, Ph.D., Ryan Hobbs Ph.D., and their respective laboratories for helpful discussions, advice, and technical expertise. We thank HG Wang, M.D., Ph.D., for providing resources for the scratch assay and Xiang Zhan, Ph.D., for statistical expertise and support. This work was funded by the NIH awards K01AR069721 (to AM Nelson) and F30ES030260 (to AM Schneider), a Medical Student Research Grant from the American Skin Association (to AM Schneider), the Department of Dermatology Research Endowment (to AM Nelson), and The Jake Gittlen Laboratories for Cancer Research at Penn State College of Medicine (to AM Nelson and AM Schneider).

### Author Contributions

AM Schneider: conceptualization, data curation, formal analysis, investigation, visualization, methodology, and writing—original draft, review, and editing.
RP Feehan: formal analysis, investigation, visualization, methodology, and writing—original draft, review, and editing.
ML Sennett: formal analysis, investigation, and writing—original draft, review, and editing.
CA Wills: formal analysis, investigation, and writing—review and editing.
C Garner: formal analysis, investigation, and writing—review and editing.
Z Cong: formal analysis, investigation, and writing—review and editing.
EM Billingsley: resources, investigation, and writing—review and editing.
AF Flamm: resources, formal analysis, and writing—review and editing.
LM Shantz: formal analysis, methodology, and writing—original draft, review, and editing.
AM Nelson: conceptualization, data curation, formal analysis, supervision, funding acquisition, methodology, project administration, and writing—original draft, review, and editing.

**Life Science Alliance**

**Conflict of Interest Statement**

The authors declare that they have no conflict of interest.

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
