## [Reviewer comments · Life Science Alliance]

TLR3 activation mediates partial epithelial to mesenchymal transition in human keratinocytes

Andrea M Schneider, Robert P Feehan, Mackenzie L Sennett, Carson A Wills, Charlotte Garner, Zhaoyuan Cong, Elizabeth M Billingsley, Alexandra F Flamm, Lisa M Shantz, Amanda M Nelson

DOI: 10.26508/lsa.202402777

Corresponding author(s): Prof. Amanda M Nelson (Penn State Milton S. Hershey Medical Center)

Review timeline:

Submission Date:	2024-04-18
Editorial Decision:	2024-06-14
Revision Received:	2024-09-11
Editorial Decision:	2024-09-12
Revision Received:	2024-09-23
Accepted:	2024-09-23

Scientific Editor: Eric Sawey

Transaction Report:

No Peer Review Process File is available with this article, as the authors have chosen not to make the review process public in this case.

Re: Life Science Alliance manuscript #LSA-2024-02777-T
Prof. Amanda M Nelson
Penn State Milton S. Hershey Medical Center
Dermatology
500 UNIVERSITY DRIVE
C7801 BMR
HERSHEY, PA 17033

Dear Dr. Nelson,

Thank you for submitting your manuscript entitled "TLR3 activation mediates partial epithelial to mesenchymal transition in human keratinocytes" to Life Science Alliance. The manuscript was assessed by expert reviewers, whose comments are appended to this letter. We invite you to submit a revised manuscript addressing the Reviewer comments.

Thank you for this interesting contribution to Life Science Alliance. We are looking forward to receiving your revised manuscript.

Sincerely,

Eric Sawey, PhD

Executive Editor

Life Science Alliance

<http://www.lsajournal.org>

B. MANUSCRIPT ORGANIZATION AND FORMATTING:

RE: Life Science Alliance Manuscript #LSA-2024-02777-TR

Prof. Amanda M Nelson
Penn State Milton S. Hershey Medical Center
Dermatology
500 UNIVERSITY DRIVE
C7801 BMR
HERSHEY, PA 17033

Dear Dr. Nelson,

Thank you for submitting your revised manuscript entitled "TLR3 activation mediates partial epithelial to mesenchymal transition in human keratinocytes". We would be happy to publish your paper in Life Science Alliance pending final revisions necessary to meet our formatting guidelines.

- please be sure that the authorship listing and order is correct
- please upload your supplemental tables as single files. Please make sure the table files are in editable doc or excel format.
- please add ORCID ID for corresponding author to our system-you should have received instructions on how to do so. Thank you for providing the ORCID ID's for your co-authors. In order for the ORCID ID to be connected, each co-author has to login to their LSA profile and connect their ORCID ID. Please get in touch with us if you experience any issues.
- please add the Twitter handle of your host institute/organization as well as your own or/and one of the authors in our system
- please use the [10 author names, et al.] format in your references (i.e. limit the author names to the first 10)
- please add a figure legend section to your main manuscript, including the main and supplemental figure legends
- please make sure the deposited RNA-seq data is publicly accessible at this time

LSA now encourages authors to provide a 30-60 second video where the study is briefly explained. We will use these videos on social media to promote the published paper and the presenting author (for examples, see <https://docs.google.com/document/d/1-UWCfbE4pGcDdcgzcmiuJI2XMBJnxKYeqRvLLrLS08s/edit?usp=sharing>). Corresponding or first-authors are welcome to submit the video. Please submit only one video per manuscript. The video can be emailed to contact@life-science-alliance.org

A. FINAL FILES:

B. MANUSCRIPT ORGANIZATION AND FORMATTING:

Sincerely,

3rd Editorial Decision

23 September 2024

RE: Life Science Alliance Manuscript #LSA-2024-02777-TRR

Prof. Amanda M Nelson
Penn State Milton S. Hershey Medical Center
Dermatology
500 UNIVERSITY DRIVE
C7801 BMR
HERSHEY, PA 17033

Dear Dr. Nelson,

Thank you for submitting your Research Article entitled "TLR3 activation mediates partial epithelial to mesenchymal transition in human keratinocytes". It is a pleasure to let you know that your manuscript is now accepted for publication in Life Science Alliance. Congratulations on this interesting work.

DISTRIBUTION OF MATERIALS:

Again, congratulations on a very nice paper. I hope you found the review process to be constructive and are pleased with how the manuscript was handled editorially. We look forward to future exciting submissions from your lab.

Sincerely,
